# Avoiding Catastrophic States with Intrinsic Fear

## Abstract

Many practical reinforcement learning problems contain catastrophic states that the optimal policy visits infrequently or never. Even on toy problems, deep reinforcement learners periodically revisit these states, once they are forgotten under a new policy. In this paper, we introduce *intrinsic fear*, a learned *reward shaping* that accelerates deep reinforcement learning and guards oscillating policies against periodic catastrophes. Our approach incorporates a second model trained via supervised learning to predict the probability of imminent catastrophe. This score acts as a penalty on the Q-learning objective. Our theoretical analysis demonstrates that the perturbed objective yields the same average return under strong assumptions and an $\epsilon$-close average return under weaker assumptions. Our analysis also shows robustness to classification errors. Equipped with intrinsic fear, our DQNs solve the toy environments and improve on the Atari games Seaquest, Asteroids, and Freeway.

## 1 Introduction

Following success on Atari games (Mnih et al., 2015) and the board game Go (Silver et al., 2016), many researchers have begun exploring practical applications of deep reinforcement learning (DRL). Some investigated applications include robotics (Levine et al., 2016), dialogue systems (Fatemi et al., 2016; Lipton et al., 2016), energy management (Night, 2016), and self-driving cars (Shalev-Shwartz et al., 2016). Amid this push to apply DRL, we might ask, *can we trust these agents in the wild?* Agents acting in real-world environments might possess the ability to cause catastrophic outcomes. Consider a self-driving car that might hit pedestrians or a domestic robot that might injure a child. We might hope to prevent DRL agents from ever making catastrophic mistakes. But doing so requires extensive prior knowledge of the environment in order to constrain the exploration of policy space (García and Fernández, 2015).

Many conflicting definitions of safety and catastrophe exist, a problem that invites further philosophical consideration. In this paper, we introduce a specific but plausible notion of *avoidable catastrophes*. These are states that prior knowledge dictates an optimal policy should never visit. For example, we might believe that an optimal self-driving algorithm would never hit a pedestrian. Moreover, we assume that an optimal policy never even comes *near* an avoidable catastrophe state. We define proximity in trajectory space, and not by the geometry of feature space. We denote states proximal to avoidable catastrophes as *danger states*. While we don't assume prior knowledge of which states are dangerous, we do assume the existence of a *catastrophe detector*. After encountering a catastrophic state, an agent can realize this and take action to avoid dangerous states in the future.

Given this definition, we address two challenges: First, can we expect DRL agents, after experiencing some number of catastrophic failures, to avoid perpetually making the same mistakes? Second, can we use our prior knowledge that catastrophes should be kept at a distance to accelerate learning of a DRL agent? Our experiments show that even on toy problems, the deep Q-network (DQN), a basic algorithm behind many of today's state-of-the-art DRL systems, struggles on both counts. Even in toy environments, DQNs may encounter thousands of catastrophes before learning to avoid them and are susceptible to repeating old errors. We call this latter problem *the Sisyphean curse*.

This poses a formidable obstacle to using DQNs in the real world. How can we hand over responsibility for consequential actions (control of a car, say) to a DRL agent if it may be doomed to periodically remake every kind of mistake, however grave, so long as it continues to learn? Imagine a self-driving

car that had to periodically hit a few pedestrians in order to remember that is undesirable. In the tabular setting, an RL agent never forgets the learned dynamics of its environment, even as its policy evolves. Moreover, if the Markovian assumption holds, eventual convergence to a globally optimal policy is guaranteed. Unfortunately, the tabular approach becomes infeasible in high-dimensional, continuous state spaces.

The trouble for DQNs owes to the use of function approximation (Murata and Ozawa, 2005). When training a DQN, we successively update a neural network based on experiences. These experiences might be sampled in an online fashion, from a trailing window (*experience replay buffer*), or uniformly from all past experiences. Regardless of which mode we use to train the network, eventually, states that a learned policy never encounters will come to form an infinitesimally small region of the training distribution. At such times, our networks are subject to the classic problem of catastrophic interference (McCloskey and Cohen, 1989; McClelland et al., 1995). Nothing prevents the DQN's policy from drifting back towards a policy that revisits long-forgotten catastrophic mistakes.

More formally, we characterize the problem as unfolding in the following steps: (i) Training under distribution $\mathcal{D}$, our agent produces a safe policy $\pi_s$ that avoids catastrophes (ii) Collecting data generated under $\pi_s$ yields a new distribution of transitions $\mathcal{D}'$ (iii) Training under $\mathcal{D}'$, the agent produces $\pi_d$, a policy that once again experiences avoidable catastrophes. To illustrate the brittleness of modern DRL algorithms, we introduce a simple pathological problem called *Adventure Seeker*. This problem consists of a one-dimensional continuous state, two actions, simple dynamics, and a clear analytic solution. Nevertheless, the DQN fails. We then show that similar dynamics exist in the classic RL environment Cart-Pole.

**In this paper**, to combat these problems, we propose *intrinsic fear*. In this approach, we train a supervised *fear model* that predicts which states are likely to lead to a catastrophe within $k_r$ steps. The output of the fear model (a probability), scaled by a *fear factor* penalizes the $Q$-learning target. Our approach draws inspiration from intrinsic motivation (Chentanez et al., 2004). However, instead of perturbing the reward function to encourage the discovery of novel states, we perturb it to discourage revisiting catastrophic states.

We validate the approach both empirically and theoretically. Our experiments address both our *Adventure Seeker* problem and Cartpole as well as the Atari games Seaquest and Asteroids, and Freeway. For these environments, we label each loss of a *life* as a catastrophic state. On the toy environments, the intrinsic fear agent learns to avoid death indefinitely, achieving unbounded reward per episode. On Seaquest and Asteroids, the intrinsic fear agent improves markedly and on Freeway the improvement is dramatic. Theoretically, we demonstrate the following: First, we prove that when the reward is bounded and the optimal policy rarely visits the catastrophic states, the policy learned on the altered value function has return similar to the optimal policy on the original value function. Second we prove that the method is robust to noise in the danger model.

## 2 INTRINSIC FEAR

Over a series of turns, an agent interacts with its environment via a Markov decision process, or MDP, $(\mathcal{S}, \mathcal{A}, \mathcal{T}, \mathcal{R}, \gamma)$. At each step $t$, an agent observes a state $s \in \mathcal{S}$. The agent then chooses an action $a \in \mathcal{A}$ according to some policy $\pi$. In turn, the environment transitions to a new state $s_{t+1} \in \mathcal{S}$ according to transition dynamics $\mathcal{T}(s_{t+1}|s_t, a_t)$ and generates a reward $r_t$ with expectation $\mathcal{R}(s, a)$. This cycle continues until each episode terminates.

The goal of an agent is to maximize the cumulative discounted return $\sum_{t=0}^{T} \gamma^t r_t$. Temporal-differences (TD) methods (Sutton, 1988) such as Q-learning (Watkins and Dayan, 1992) model the Q-function, which gives the *optimal* discounted total reward of a state-action pair; the greedy policy w.r.t. the Q-function is optimal (Sutton and Barto, 1998). Problems of practical interest tend to have large state spaces, thus the Q-function is typically approximated by parametric models such as neural networks.

In Q-learning with function approximation, an agent alternately collects experiences by acting greedily with respect to $Q(s, a; \theta_Q)$ and updates its parameters $\theta_Q$. Updates proceed as follows. For a given experiences $(s_t, a_t, r_t, s_{t+1})$, we minimize the squared Bellman error:

$$\mathcal{L} = (Q(s_t, a_t; \theta_Q) - y_t)^2 \tag{1}$$

for $y_t = r_t + \gamma \cdot \max_{a'} Q(s_{t+1}, a'; \theta_Q)$. Traditionally, the parameterised $Q(s, a; \theta)$ is trained by stochastic approximation, estimating the loss on each experience as it is encountered, yielding the update:

$$\theta_{t+1} \leftarrow \theta_t + \alpha(y_t - Q(s_t, a_t; \theta_t)) \nabla Q(s_t, a_t; \theta_t).$$ (2)

Q-learning methods also require an exploration strategy for action selection. For simplicity, we consider only the $\epsilon$-greedy heuristic.

A few tricks help to stabilize Q-learning with function approximation. Of particular relevance to this work is experience replay (Lin, 1992): the RL agent maintains a buffer of past experiences, applying TD-learning on randomly selected mini-batches of experience to update the Q-function.

In this paper, we propose a new formulation of the safety problem. We suppose there exists a subset $\mathcal{C} \subset \mathcal{S}$ of states that an optimal policy encounters them very rarely or never and denote them *catastrophic states*. Moreover, we assume that for some environments, optimal policies are rarely within a short distance of a catastrophic state. As a measure of distance, we consider steps in trajectory space. We define the distance $d(s_i, s_j)$ to be length $N$ of the smallest sequence of transitions $\{(s_t, a_t, r_t, s_{t+1})\}_{t=1}^{N}$ that traverses state space from $s_i$ to $s_j$.[1]

**Definition 2.1.** Suppose that we are given a priori knowledge that acting according to the optimal policy $\pi^*$, an agent never encounters states $s \in S$ for which lie within distance $d(s, c) < k_\tau$ for any catastrophe state $c \in \mathcal{C}$. Then each state $s$ for which $\exists c \in \mathcal{C}$ s.t. $d(s, c) < k_\tau$ is a *danger state*.

We also suppose that the agent can recognize the catastrophe states as they are encountered.

**Definition 2.2.** A *catastrophe detector* is a function $f : \mathcal{S} \mapsto \{0, 1\}$ that returns 1 if and only if a state is a catastrophe state.

We propose Intrinsic Fear (IF) (Algorithm 1), a novel algorithm for avoiding catastrophes when learning online with function approximation. In our approach, we maintain both a DQN and a separate, supervised *fear model* $F : \mathcal{S} \mapsto [0, 1]$. Our fear model $F$ provides an auxiliary source of reward, penalizing the Q-learner for entering possibly dangerous states.

The goal in modeling danger states is twofold. First, by shaping rewards away from suboptimal states, we encode prior knowledge about the environment and can thus accelerates learning. Second, when catastrophic states correspond to especially undesirable outcomes, the learned reward shaping can protect DQNs, which are susceptible to catastrophic forgetting, from drifting close to catastrophic states. Owing to this self-assigned reward, once the fear model is trained, a Q-learner might update to avoid catastrophes without having to actually repeat them, so long as the fear model is not itself susceptible to catastrophic forgetting. We draw some inspiration from the idea of a parent scolding a child for running around with a knife. The child can learn to adjust its behavior without actually having to stab someone. We also draw inspiration from the way humans appear to process traumatic experience, remembering especially bad events vividly even as most other memories from the same time period fade. Perhaps this selective memorization of bad events confers a benefit for avoiding similar outcomes in the future.

Our instantiation of intrinsic fear works as follows: In addition to the DQN, we maintain a binary classifier that we term a *fear model*. In our case, we use a neural network of the same architecture as the DQN (but for the output layer). The fear model's purpose is to predict the probability that any state will lead to catastrophe within $k$ moves. Over the course of training, our agent adds each experience $(s, a, r, s')$ to its experience replay buffer. As each catastrophe is reached at the $n_{th}$ turn of an episode, we add the $k_r$ (*fear radius*) states leading up to the catastrophe to a list of *danger states*. We add the preceding $n - k_r$ states to a list of *safe states*. When $n < k_r$, all states for that episode are added to the list of danger states. Then after each turn, in addition to making one update to the Q-network, we make one mini-batch update to the fear model. To make this update, we sample $50\%$ of samples in the batch from the *danger states*, assigning them label 1 and the remaining $50\%$ from the *safe states*, assigning them label 0.

For each update to the DQN, we perturb the TD target $y_t$. Instead of updating $Q(s_t, a_t; \theta_Q)$ towards $r_t + \max_{a'} Q(s_{t+1}, a'; \theta_Q)$, we introduce the *intrinsic fear* to the model via the target:

$$y_t^{IF} = r_t + \max_{a'} Q(s_{t+1}, a'; \theta_Q) - \lambda \cdot F(s_{t+1}; \theta_F)$$ (3)

---

[1]In the stochastic dynamics setting, the distance is the minimum mean passing time between the states.

---

**Algorithm 1** Training DQN with Intrinsic Fear

---

1: **Input:** Two models: $Q$ (DQN) and $F$ (fear model), fear factor $\lambda$, fear phase-in length $k_\lambda$, fear radius $k_r$
2: **Output:** Learned parameters $\theta_Q$ and $\theta_F$
3: Initialize parameters $\theta_Q$ and $\theta_F$ randomly
4: Initialize replay buffer $\mathcal{D}$, danger state buffer $\mathcal{D}_D$, and safe state buffer $\mathcal{D}_S$
5: Start per-episode turn counter $n_e$
6: **for** $t$ in 1:$T$ **do**
7:     With probability $\epsilon$ select random action $a_t$
8:     Otherwise, select action $a_t = argmax_{a'} Q(s_t, a'; \theta_Q)$
9:     Execute action $a_t$ in environment, observing reward $r_t$ and successor state $s_{t+1}$
10:     Store transition $(s_t, a_t, r_t, s_{t+1})$ in $\mathcal{D}$
11:     **if** $s_{t+1}$ is a catastrophe state **then**
12:         Add states $s_{t-k_r}$ through $s_t$ to $\mathcal{D}_D$
13:     **else**
14:         Add states $s_{t-n_e}$ through $s_{t-k_r-1}$ to $\mathcal{D}_S$
15:     Sample random minibatch of transitions $(s_\tau, a_\tau, r_\tau, s_{\tau+1})$ from $\mathcal{D}$
16:     $\lambda_\tau \leftarrow \min(\lambda, \frac{\lambda \cdot t}{k_\lambda})$
17:     $y_\tau \leftarrow \begin{cases} r_\tau - \lambda_\tau, & \text{for terminal } s_{\tau+1} \\ r_\tau + \max_{a'} Q(s_{\tau+1}, a'; \theta_Q) - \lambda \cdot F(s_{\tau+1}; \theta_F) & \text{for non-terminal } s_{\tau+1} \end{cases}$
18:     $\theta_Q \leftarrow \theta_Q - \eta \cdot \nabla_{\theta_Q}(y_\tau - Q(s_\tau, a_\tau; \theta_Q))^2$
19:     Sample random mini-batch $s_j$ with 50% of examples from $\mathcal{D}_D$ and 50% from $\mathcal{D}_S$
20:     $y_j \leftarrow \begin{cases} 1, & \text{for } s_j \in \mathcal{D}_D \\ 0, & \text{for } s_j \in \mathcal{D}_S \end{cases}$
21:     $\theta_F \leftarrow \theta_F - \eta \cdot \nabla_{\theta_F} \text{loss}_F(y_j, F(s_j; \theta_F))$

---

where $F(s; \theta_F)$ is the fear model and $\lambda$ is a *fear factor* determining the scale of the impact of intrinsic fear on the Q-function update.

Note that IF perturbs the objective function. Thus, one might be concerned that the perturbed reward might indicate a different optimal policy. Fortunately, if the labeled catastrophe states and danger zone do not violate our assumptions, and if the fear model reaches arbitrarily high accuracy, then this will not happen.

For an MDP, $M = \langle \mathcal{S}, \mathcal{A}, \mathcal{T}, \mathcal{R}, \gamma \rangle$, with $0 \le \gamma \le 1$, the average reward return is as follows:

$$\eta_M(\pi) := \begin{cases} \lim_{T \to \infty} \frac{1}{T} \mathbb{E}_M \left[ \sum_t^T r_t | \pi \right] & \text{if } \gamma = 1 \\ (1 - \gamma) \mathbb{E}_M \left[ \sum_t^\infty \gamma^t r_t | \pi \right] & \text{if } 0 \le \gamma < 1 \end{cases} \tag{4}$$

The optimal policy $\pi^*$ of the model $M$ is the policy which maximizes the average reward return, $\pi^* = \max_{\pi \in \mathcal{P}} \eta(\pi)$ where $\mathcal{P}$ is a set of stationary polices.

**Theorem 1.** *For a given MDP, $M$, with $\gamma \in [0,1]$ and a catastrophe detector $f$, let $\pi^*$ denote an optimal policy of $M$, and $\tilde{\pi}$ denote an optimal policy of $M$ equipped with fear model $F$ and $\lambda$. If the probability $\pi^*$ visits the states in the danger zone is at most $\epsilon$, and $\mathcal{R}_{\min} \le \mathcal{R}(s, a) \le \mathcal{R}_{\max}$, then*

$$\eta_M^* \ge \eta_M(\tilde{\pi}) \ge \eta_{M,F}(\tilde{\pi}) \ge \eta_M^* - \lambda \epsilon (\mathcal{R}_{\max} - \mathcal{R}_{\min}) . \tag{5}$$

*Proof.* Appendix A.

It is worth noting that when at least one of the optimal policies of $M$, does not visit the fear zone ($\epsilon = 0$), then $\eta_M^* = \eta_{M,F}(\tilde{\pi})$ and the fear signal can boost up the process of learning the optimal policy.

Since we learn the catastrophe detector $f$ and fear model $F$ empirically using the collected data, our RL agent has access to an imperfect detector $\hat{f}$ and imperfect fear model $\hat{F}$, and therefore assumes the fear model is $\hat{F}$. In this case, the RL agent trains with intrinsic fear generated by $\hat{f}$, learning a different value function than the RL agent with perfect $f$. To show robustness against modeling errors, we are interested in the average deviation in the value functions of the two agents.

In general, in practical RL problems, we use discount factors $\gamma < 1$ (Kocsis and Szepesvári, 2006) in order to reduce the planing horizon, and computation cost. Moreover, (Jiang et al., 2015) suggests that when we have estimation (up to the confidence intervals) of our MDP model, it is better to use smaller discount factors in order to prevent over-fitting to the estimated model. We show that under modeling errors, if the actual objective function to optimize for Eq. 4 has with discount factor $\gamma_{eval}$, it's better to use some $\gamma \leq \gamma_{eval}$ because it reduces the average deviation in the value functions.

For a given environment, with fear model $F_1$ and discount factor $\gamma_1$, let $V_{F_1,\gamma_1}^{\pi_{F_2,\gamma_2}^*}(s)$, $s \in \mathcal{S}$, denote the state value function under the optimal policy of a environment with fear model $F_2$ and the discount factor $\gamma_2$. On the same environment, let $\omega_{F_1}^{\pi_{F_2,\gamma_2}^*}(s)$ denote the stationary distribution over states. Therefore we are interested in the average deviation on value functions caused by imperfect classifier:

$$\mathcal{L}(F, \widehat{F}, \gamma_{eval}, \gamma) := (1 - \gamma_{eval}) \int_{s \in \mathcal{S}} \omega_F^{\pi_{\widehat{F},\gamma}^*}(s) \left| V_{F,\gamma_{eval}}^{\pi_{F,\gamma_{eval}}^*}(s) - V_{F,\gamma_{eval}}^{\pi_{\widehat{F},\gamma}^*}(s) \right| ds$$

**Theorem 2.** *For a given MDP model, the average deviation on the value functions, $\mathcal{L}(F, \widehat{F}, \gamma_{eval}, \gamma)$, $F, \hat{F} \in \mathcal{F}$, vanishes as the number of samples $N$ increases*

$$\mathcal{L} = \mathcal{O}\left(\lambda(\mathcal{R}_{\max} - \mathcal{R}_{\min}) \frac{1 - \gamma_{eval}}{1 - \gamma} \frac{\mathcal{VC}(\mathcal{F}) + \log \frac{1}{\delta}}{N} + \frac{\gamma_{eval} - \gamma}{1 - \gamma}\right) \tag{6}$$

*with probability at least $1 - \delta$. $\mathcal{VC}(\mathcal{F})$ is the $\mathcal{VC}$ dimension of the hypothesis class $\mathcal{F}$.*

*Proof.* Appendix B

Thm. 2, holds for both tabular MDPs and continuous state-action MDPs. In addition to proofs of these results, we provide a deeper theoretical analysis on deterministic and stochastic fear models in the tabular setting in Appendix B.

Over the course of our experiments, we discovered the following pattern: Intrinsic fear models are more effective when the *fear radius* $k_r$ is large enough that the model can experience danger states at a safe distance and correct the policy, without experiencing many catastrophes. When the fear radius is too small, the danger probability is only nonzero at states from which catastrophes are inevitable anyway and intrinsic fear seems not to help. We also found that wider fear factors train more stably when phased in over the course of many episodes. So, in all of our experiments we gradually phase in the *fear factor* $\lambda$ from 0 to $\lambda$ reaching full strength at predetermined time step $k_\lambda$. In our Cart-Pole experiments, we phase $\lambda$ in over $1M$ steps.

## 3 ENVIRONMENTS

We demonstrate our algorithms on three environments. These include *Adventure Seeker*, a toy pathological environments which we designed to demonstrate the Sisyphean curse; *Cartpole*, a classic reinforcement learning environment; and three Atari games, *Seaquest*, *Asteroids*, and *Freeway*, simulated in the Arcade Learning Environment (Bellemare et al., 2013).

**Adventure Seeker** We imagine a player placed on a hill, sloping upward to the right (Figure 1a). At each turn, the player can move to the right (up the hill) or left (down the hill). The environment adjusts the player's position accordingly, adding some random noise. Between the left and right edges of the hill, the player gets more reward for spending time higher on the hill. But if the player goes too far to the right, he/she will fall off (a *catastrophic state*), terminating the episode and receiving a return of 0. Formally, the state consists of a single continuous variable $s \in [0, 1.0]$, denoting the player's position. The starting position for each episode is chosen uniformly at random in the interval $[.25, .75]$. The available actions consist only of $\{-1, +1\}$ (*left* and *right*). Given an action $a_t$ in

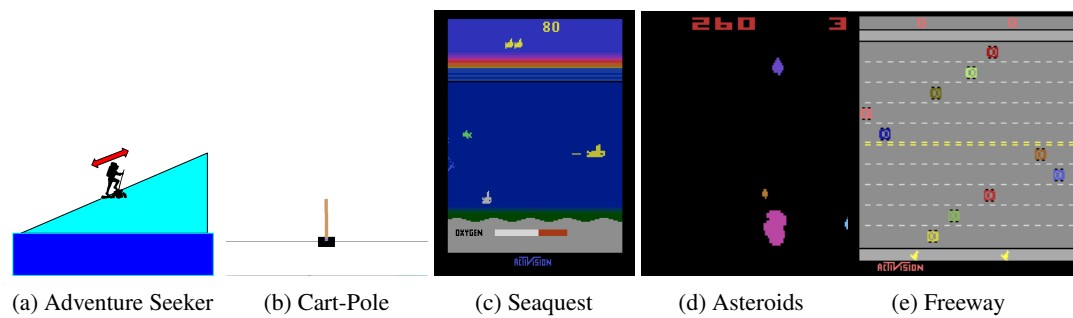

(a) Adventure Seeker     (b) Cart-Pole     (c) Seaquest     (d) Asteroids     (e) Freeway

Figure 1: In experiments, we consider two toy environments (a,b) and the Atari games Seaquest (c), Asteroids (d), and Freeway (e)

state $s_t$, $\mathcal{T}(s_{t+1}|s_t, a_t)$ gives successor state $s_{t+1} \leftarrow s_t + .01 \cdot a_t + \eta$ where $\eta \sim \mathcal{N}(0, .01^2)$. The reward at each turn is equal to $s_t$ (proportional to height). The player falls off the hill, entering the catastrophic terminating state, whenever $s_{t+1} > 1.0$ or $s_{t+1} < 0.0$.

This game admits an obvious analytic solution; There exists some threshold above which the agent should always choose to go left, and below which it should always go right. And yet a state-of-the-art DQN model learning online or with experience replay successively plunges to its death. To be clear, the DQN does learn a near-optimal thresholding policy quickly. But over the course of continued training, the agent oscillates between a reasonable thresholding policy and one which always moves right, regardless of the state. The pace of this oscillation evens out and all networks (over multiple runs) quickly reach a constant catastrophe per turn rate that does not attenuate with continued training. How could we trust a system that can't solve *Adventure Seeker* to make consequential decisions?

**Cart-Pole** In this classic RL environment, an agent balances a pole atop a cart (Figure 1b). Qualitatively, the game exhibits four distinct catastrophe modes. The pole could fall down to the right or fall down to the left. Additionally, the cart could run off the right boundary of the screen or run off the left. Formally, at each time, the agent observes a four-dimensional state vector $(x, v, \theta, \omega)$ consisting respectively of the cart position, cart velocity, pole angle, and the pole's angular velocity. At each time step, the agent chooses an action, applying a force of either $-1$ or $+1$. For every time step that the pole remains upright and the cart remains on the screen, the agent receives a reward of $1$. If the pole falls, the episode terminates, giving a return of $0$ from the penultimate state. In experiments, we use the implementation *CartPole-v0* contained in the openAI gym (Brockman et al., 2016). Like Adventure Seeker, this problem admits an analytic solution. A perfect policy should never drop the pole. But, as with Adventure Seeker, a DQN converges to a constant rate of catastrophes per turn.

**Atari games** In addition to these pathological cases, we address Freeway, Asteroids, and Seaquest, games from the Atari Learning Environment. In Freeway, the agent controls a chicken with a goal of crossing the road while dodging traffic. The chicken loses a life and starts from the original location if hit by a car. Points are only rewarded for successfully crossing the road. In Asteroids, the agent pilots a ship and gains points from shooting the asteroids. She must avoid colliding with asteroids which cost it lives. In Seaquest, a player swims under water. Periodically, as the oxygen gets low, she must rise to the surface for oxygen. Additionally, fishes swim across the screen. The player gains points each time she shoots a fish. Colliding with a fish or running out of oxygen result in death. In all three games, the agent has 3 lives, and the final death is a terminal state. We label each loss of a life as a catastrophe state.

## 4 EXPERIMENTS

To assess the effectiveness of the intrinsic fear model, we evaluate both a standard DQN (DQN-NoFear) and one enhanced by *intrinsic fear* (DQN-Fear). In both cases, we use multilayer perceptrons (MLPs) with a single hidden layer and $128$ hidden nodes. We train all MLPs by stochastic gradient descent using the Adam optimizer Kingma and Ba (2015) to adaptively tune the learning rate.

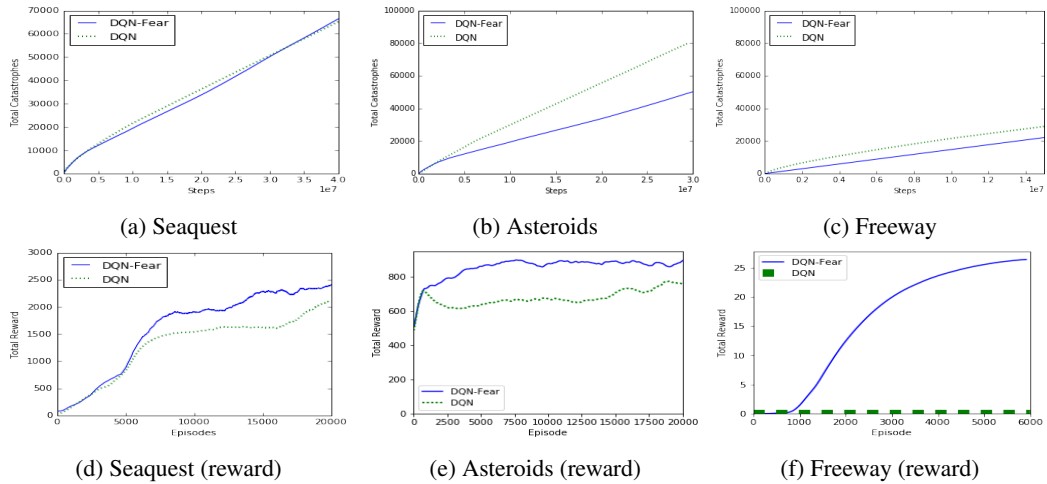

Figure 2: Catastrophes and reward/episode for DQNs and *Intrinsic Fear*. On Adventure Seeker, all Intrinsic Fear models cease to "die" within 14 runs, giving unbounded (unplottable) reward thereafter. On Seaquest, the IF model achieves a similar catastrophe rate but significantly higher total reward. On Asteroids, the IF model outperforms DQN. For Freeway, a randomly exploring DQN (under our time limit) never gets reward but IF model learns successfully.

Because, for the *Adventure Seeker* problem, an agent can escape from danger with only a few time steps of notice, we set the fear radius $k_r$ to 5. We phase in the fear factor quickly, reaching full strength in just 1000 moves. On this problem we set the fear factor $\lambda$ to 40.

For *Cart-Pole*, we set a wider fear radius of $k_r = 20$. We initially tried training this model with a shorter fear radius but made the following observation. Some models would learn well surviving for millions of experiences, with just a few hundred catastrophes. This compared to a DQN (Figure 2) which would typically suffer 4000-5000 catastrophes. When examining the output from the fear models on successful vs unsuccessful runs, we noticed that the unsuccessful models would output danger of probability greater than .5 for precisely the 5 moves before a catastrophe. But by that time it would be too late for an agent to correct course. In contrast, on the more successful runs, the fear model typically outputs predictions in the range $.1 - .5$. We suspect that the gradation between mildly dangerous states and those with imminent danger provides a richer reward signal to the DQN.

On both the Adventure Seeker and Cart-Pole environments, the DQNs augmented by intrinsic fear far outperform their otherwise identical counterparts (Figure 2). We cannot plot the reward per episode for the intrinsic fear models on these environments because after the first several deaths, the episodes never terminate. In contrast, both the DQN and related approaches like expected SARSA continue to visit the catastrophic states regularly. We compared our approach against some traditional approaches for mitigating catastrophic forgetting. For example, we tried a memory-based method in which we preferentially sample the catastrophic states for updating the model, but they did not improve over the DQN. It seems that the notion of a danger zone is necessary here.

For Seaquest, Asteroids, and Freeway, we use a fear radius of 5 and a fear factor of .5. For all Atari games, the IF models outperform their DQN counterparts. Interestingly while for all games, the IF models achieve higher reward, on Seaquest, models trained with Intrinsic Fear have similar catastrophe rates. More precisely, they appear to have fewer catastrophes early on but eventually enter a different reward regime, exchanging more catastrophes for higher reward. This result suggests an interplay between the various reward signals that warrants further exploration. For Asteroids and Freeway, the improvements are more dramatic. Over just a few thousand episodes of Freeway, a randomly exploring DQN achieves zero reward. However, the reward shaping of intrinsic fear leads to rapid improvement.

## 5 RELATED WORK

The paper addresses safety in RL, intrinsically motivated RL, and the stability of Q-learning with function approximation under distributional shift. Our work also has some connection to reward

shaping. We attempt to highlight the most relevant papers here. Several papers address safety in RL. (García and Fernández, 2015) provide a thorough review on the topic, identifying two main classes of methods: those that perturb the objective function and those that use external knowledge to improve the safety of exploration.

While a typical reinforcement learner optimizes expected return, some papers suggest that a safely acting agent should also minimize risk. (Hans et al., 2008) defines a *fatality* as any return below some threshold $\tau$. They propose a solution comprised of a *safety function*, which identifies unsafe states, and a *backup model*, which navigates away from those states. Their work, which only addresses the tabular setting, suggests that an agent should minimize the probability of fatality instead of maximizing the expected return. Heger (1994) suggests an alternative Q-learning objective concerned with the minimum (vs expected) return. Other papers suggest modifying the objective to penalize policies with high-variance returns (García and Fernández, 2015). Maximizing expected returns while minimizing their variance is a classic problem in finance, where a common objective is the ratio of expected return to its standard deviation (Sharpe, 1966). (Moldovan and Abbeel, 2012) gives a definition of safety based on ergodicity. They consider a fatality to be a state from which one cannot return to the start state. Shalev-Shwartz et al. (2016) theoretically analyzes how strong a penalty should be to discourage accidents. They also consider hard constraints to ensure safety. None of the above works address the case where distributional shift dooms an agent to perpetually revisit known catastrophic failure modes. Other papers incorporate external knowledge into the exploration process. Typically, this requires access to an oracle or extensive prior knowledge of the environment. In the extreme case, some papers suggest confining the policy search to the subset of policies known to be *safe*. For reasonably complex environments or classes of policies this seems infeasible.

The potential oscillatory or divergent behavior of Q-learners with function approximation has been previously identified (Boyan and Moore, 1995; Baird et al., 1995; Gordon, 1996). Outside of RL, the problem of covariate shift has been extensively studied (Sugiyama and Kawanabe, 2012). Murata and Ozawa (2005) addresses the problem of catastrophic forgetting owing to distributional shift in RL with function approximation, proposing a memory-based solution. Many papers address intrinsic rewards, which are internally assigned, vs the standard (extrinsic) reward. Typically, intrinsic rewards are used to encourage exploration (Schmidhuber, 1991; Bellemare et al., 2016) and to acquire a modular set of skills (Chentanez et al., 2004). Some papers refer to the intrinsic reward for discovery as *curiosity*. Like classic work on intrinsic motivation, our methods perturb the reward function. But instead of assigning bonuses to encourage discovery of novel transitions, we assign penalties to discourage catastrophic transitions.

**Key differences**   In this paper, we undertake a novel treatment of safe reinforcement learning, While the literature offers several notions of safety in reinforcement learning, we see the following problem: Existing safety research that perturbs the reward function requires little foreknowledge, but fundamentally changes the objective globally. On the other hand, processes relying on expert knowledge may presume an unreasonable level of foreknowledge. Moreover, little of the prior work on safe reinforcement learning, to our knowledge, specifically addresses the problem of catastrophic forgetting. This paper proposes a new class of algorithms for avoiding catastrophic states and a theoretical analysis supporting its robustness.

# 6   CONCLUSIONS

Our experiments demonstrate that DQNs are susceptible to periodically repeating mistakes, however bad, raising questions about their real-world utility when harm can come of actions. While it's easy to visualize these problems on toy examples, similar dynamics are embedded in more complex domains. Consider a domestic robot acting as a barber. The robot might receive positive feedback for giving a closer shave. This reward encourages closer contact at a steeper angle. Of course, the shape of this reward function belies the catastrophe lurking just past the optimal shave. Similar dynamics might be imagines in a vehicle that is rewarded for traveling faster but could risk an accident with excessive speed. Our results with the intrinsic fear model suggest that with only a small amount of prior knowledge (the ability to recognize catastrophe states after the fact), we can simultaneously accelerate learning and avoid catastrophic states. This work represents a first step towards combating some issues relating to safety in RL stemming from catastrophic forgetting.

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

# A  LOSS IN OPTIMAL VALUE (PROOF OF THEOREM 1)

The average return of the reward under a policy $\pi$ is as follows:

$$\eta_M(\pi) = \lim_{T\to\infty} \frac{1}{T}\mathbb{E}\left[\sum_t^T r_t|\pi\right].\tag{7}$$

Let us assume that any stationary policy $\pi$ induces a stationary distribution $\omega_\pi(s),\ s \in \mathcal{S}$. Therefore we can rewrite Eq. 7 in terms of stationary distribution (Puterman, 2014).

$$\eta_M(\pi) = \lim_{T\to\infty} \frac{1}{T}\mathbb{E}\left[\sum_t r_t|\pi\right] = \sum_{s\in\mathcal{S}}\sum_{a\in\mathcal{A}}\omega_\pi(s)\pi(a|s)\mathcal{R}(s,a).$$

In RL, we are interested in a policy $\pi^*$ that maximizes the expected average reward:

$$\pi^* := \arg\max_\pi \eta_M(\pi).$$

Denote $\eta_M^* = \eta_M(\pi^*)$. In a first place, the optimization in Eq. 7 looks linear in $\pi$ but actually the policy $\pi$ derives the stationary distribution $\omega_\pi(\cdot)$, which makes the optimization problem harder.

Given the policy $\pi$, let's define the joint distribution in $(s,a)$ as follows:

$$\mu_\pi(s,a) := \mathbb{P}(s,a|\pi) = \omega_\pi(s)\pi(a|s),\ \forall s \in \mathcal{S}, a \in \mathcal{A}.$$

Then we can rewrite the optimization problem in terms of the joint probability distribution $\mu_\pi$.

$$\eta_m(\mu_\pi) := \sum_{s\in\mathcal{S}}\sum_{a\in\mathcal{A}}\mu_\pi(s,a)\mathcal{R}(s,a).\tag{8}$$

We can see that this new formalization, turns our optimization objective into a linear function of $\mu_\pi$. Since $\mu_\pi$ is a join distribution of $(s,a)$ under the model dynamics $\mathcal{T}$, it can not take any arbitrary value. Let $\Delta$ denote the set of feasible value for $\mu_\pi$, then

$$\Delta := \{\mu : \mu \geq 0, \sum_{s,a}\mu(s,a) = 1, \sum_{a'}\mu(s',a') = \sum_{s,a}\mathcal{T}(s'|s,a)\mu(s,a), \forall s' \in \mathcal{S}\}.\tag{9}$$

Clearly, $\Delta$ is a polytope on the simplex in $\mathbb{R}^{S\times A}$.

Now, we can rewrite the optimization problem Eq. 7 as a constrained linear program on $\mu_\pi$:

$$\eta_M^* = \max_{\mu\in\Delta}\sum_{s\in\mathcal{S}}\sum_{a\in\mathcal{A}}\mu(s,a)\mathcal{R}(s,a).$$

This change of variable allows us to analyze the introduction of intrinsic fear in different situations. Among all the optimal policies of Eq. 7, consider the one which minimizes $\epsilon := \sum_{s\in\mathcal{C},a}\mu_{\pi^*}(s,a)$.

let's assume that the negative reward assigned to the states in the danger zone is $\lambda(\mathcal{R}_{\max} - \mathcal{R}_{\min})$ and the optimal policy $\tilde{\pi}$ of the environment with the intrinsic fear has return of $\eta_{M,F}(\tilde{\pi})$.

Applying policy $\pi^*$ on the environment with intrinsic fear gives a return of $\eta_M^* - \lambda\epsilon(\mathcal{R}_{\max} - \mathcal{R}_{\min})$. Therefore, the return of $\tilde{\pi}$ on the environment with intrinsic fear, $\eta_{M,F}$, is lower bounded by $\eta_M^* - \lambda\epsilon(\mathcal{R}_{\max} - \mathcal{R}_{\min})$. Therefore, applying $\tilde{\pi}$ on the environment without intrinsic fear gives a return of $\eta_M(\tilde{\pi})$ which is lower bounded by $\eta_M^* - \lambda\epsilon(\mathcal{R}_{\max} - \mathcal{R}_{\min})$.

$$\eta_M^* \geq \eta_M(\tilde{\pi}) \geq \eta_{M,F}(\tilde{\pi}) \geq \eta_M^* - \lambda\epsilon(\mathcal{R}_{\max} - \mathcal{R}_{\min}).\tag{10}$$

## A.1  DISCOUNTED CUMULATIVE REWARD

For the $\gamma$-discounted setting, we are interested in

$$\eta(\pi) = (1 - \gamma)\lim_{T\to\infty}\mathbb{E}\left[\sum_{t=0}\gamma^t r_t\right]\tag{11}$$

The above mentioned equations hold in this setting as well, *i.e.*, $\eta_M^* \geq \eta_M(\tilde{\pi}) \geq \eta_{M,F}(\tilde{\pi}) \geq \eta_M^* - \lambda\epsilon(\mathcal{R}_{\max} - \mathcal{R}_{\min})$.

# B  IMPERFECT CLASSIFIER (PROOF OF THEOREM 2)

In the previous section, we assumed that we have access to the perfect classifier $F$ which can exactly label the danger zone. This assumption does not hold in real world where we train the classifier. In this section we derive an analysis in order to show that imperfect classifier $\widehat{F}$ can not change the overall performance by much.

In general, in practical RL problems, we use discount factors $\gamma_{eval} < 1$ (Kocsis and Szepesvári, 2006) in order to reduce the planing horizon, and computation cost. Moreover, (Jiang et al., 2015) suggest that when we have an estimation (up to the confidence intervals) of our MDP model, it is better to use $\gamma \leq \gamma_{eval}$. They show that since larger discount factor enriches the class of optimal policies for a given set of plausible models, large discount factors enrich models and end up over fitting to the noisy estimate of the environment.

In this section, we show how to choose the discount factor $\gamma \leq \gamma_{eval}$ such that the learned Value function stays close to the Value function under the perfect classifier $F$ is perfect. Let, $V_{F_2,\gamma_2}^{\pi_{F_1}^*,\gamma_1}(s)$, $s \in \mathcal{S}$, denote the state value under the optimal policy of model with classifier $F_1$ under the discount factor $\gamma_1$ on the environment equipped with classifier $F$ and discount factor $\gamma_2$. On the same environment, $\omega_{F_2}^{\pi_{F_1}^*,\gamma_1}(s)$ denotes the stationary distribution over states. We are interested in the average deviation on value functions caused by the imperfect classifier:

$$\mathcal{L} := (1 - \gamma_{eval}) \sum_{s \in \mathcal{S}} \omega_F^{\pi_{\widehat{F}}^*,\gamma}(s) \left| V_{F,\gamma_{eval}}^{\pi_F^*,\gamma_{eval}}(s) - V_{F,\gamma_{eval}}^{\pi_{\widehat{F}}^*,\gamma}(s) \right|$$

This quantity can be upper bounded by

$$\mathcal{L} \leq (1 - \gamma_{eval}) \| V_{F,\gamma_{eval}}^{\pi_F^*,\gamma_{eval}} - V_{F,\gamma_{eval}}^{\pi_{\widehat{F}}^*,\gamma} \|_\infty \tag{12}$$

The goal is to find an $\gamma^*$ which minimizes this loss, i.e. $\gamma^* = argmin_{\gamma \leq \gamma_{eval}} \mathcal{L}$ with high probability. For simplicity and without loss of generality, let's assume that all the rewards, including the intrinsic fears, are in $[0, 1]$ and call $\lambda'$, the transformed version of $\lambda$ [2]. One can decompose the upper bound in Eq. 12 as follows:

$$V_{F,\gamma_{eval}}^{\pi_F^*,\gamma_{eval}}(s) - V_{F,\gamma_{eval}}^{\pi_{\widehat{F}}^*,\gamma}(s) = \left( V_{F,\gamma_{eval}}^{\pi_F^*,\gamma_{eval}}(s) - V_{F,\gamma}^{\pi_F^*,\gamma_{eval}}(s) \right) + \left( V_{F,\gamma}^{\pi_F^*,\gamma_{eval}}(s) - V_{F,\gamma_{eval}}^{\pi_{\widehat{F}}^*,\gamma}(s) \right) \tag{13}$$

The first term is the deviation on value function when applying same policy on the same environment but with different discount factors. Since $\gamma \leq \gamma_{eval}$ we have $V_{F,\gamma}^{\pi_F^*,\gamma_{eval}}(s) \leq V_{F,\gamma_{eval}}^{\pi_F^*,\gamma_{eval}}(s)$.

$$V_{F,\gamma_{eval}}^{\pi_F^*,\gamma_{eval}}(s) - V_{F,\gamma}^{\pi_F^*,\gamma_{eval}}(s) = \mathbb{E}_F \left[ \sum_{t=0}^\infty \gamma_{eval}^t r_t | s_0 = s, \pi_{F,\gamma_{eval}}^* \right] - \mathbb{E}_F \left[ \sum_{t=0}^\infty \gamma^t r_t | s_0 = s, \pi_{F,\gamma_{eval}}^* \right] \tag{14}$$

$$= \mathbb{E}_F \left[ \sum_{t=0}^\infty (\gamma_{eval}^t - \gamma^t) r_t | s_0 = s, \pi_{F,\gamma_{eval}}^* \right] \leq \left( \frac{1}{1 - \gamma_{eval}} - \frac{1}{1 - \gamma} \right) \tag{15}$$

The second part of Eq. 13 is the deviation in value function under different policies and different classifiers. Again, since $\gamma \leq \gamma_{eval}$, we have $V_{F,\gamma_{eval}}^{\pi_{\widehat{F}}^*,\gamma}(s) \geq V_{F,\gamma}^{\pi_{\widehat{F}}^*,\gamma}(s)$

$$V_{F,\gamma}^{\pi_F^*,\gamma_{eval}}(s) - V_{F,\gamma_{eval}}^{\pi_{\widehat{F}}^*,\gamma}(s) \leq V_{F,\gamma}^{\pi_F^*,\gamma_{eval}}(s) - V_{F,\gamma}^{\pi_{\widehat{F}}^*,\gamma}(s) \leq V_{F,\gamma}^{\pi_F^*,\gamma}(s) - V_{F,\gamma}^{\pi_{\widehat{F}}^*,\gamma}(s) \tag{16}$$

---

[2]Shifting and then rescaling the reward is equivalent to shifting and rescaling the Q and Value function, and does not change the optimal policy. Moreover the mentioned transformation is $r \to (r - (\mathcal{R}_{min} - \lambda(\mathcal{R}_{max} - \mathcal{R}_{min})))/((\mathcal{R}_{max} - \mathcal{R}_{min})(1 + \lambda))$, therefore, $\lambda' = \lambda/(1 + \lambda)$

where the last inequality is due to the optimality of $\pi^*_{F,\gamma}$ on the environment of $F, \gamma$. To bound this part we exploit the proof trick used in (Jiang et al., 2015).

$$V^{\pi^*_{F,\gamma}}_{F,\gamma}(s) - V^{\pi^*_{\widehat{F},\gamma}}_{F,\gamma}(s) = \left(V^{\pi^*_{F,\gamma}}_{F,\gamma}(s) - V^{\pi^*_{F,\gamma}}_{\widehat{F},\gamma}(s)\right) + \left(V^{\pi^*_{F,\gamma}}_{\widehat{F},\gamma}(s) - V^{\pi^*_{\widehat{F},\gamma}}_{\widehat{F},\gamma}(s)\right) + \left(V^{\pi^*_{\widehat{F},\gamma}}_{\widehat{F},\gamma}(s) - V^{\pi^*_{\widehat{F},\gamma}}_{F,\gamma}(s)\right) \tag{17}$$

since the middle term is negative we have

$$V^{\pi^*_{F,\gamma}}_{F,\gamma}(s) - V^{\pi^*_{\widehat{F},\gamma}}_{F,\gamma}(s) \leq \left(V^{\pi^*_{F,\gamma}}_{F,\gamma}(s) - V^{\pi^*_{F,\gamma}}_{\widehat{F},\gamma}(s)\right) + \left(V^{\pi^*_{\widehat{F},\gamma}}_{\widehat{F},\gamma}(s) - V^{\pi^*_{\widehat{F},\gamma}}_{F,\gamma}(s)\right) \leq 2 \max_{\{\pi^*_{\widehat{F},\gamma}, \pi^*_{\widehat{F},\gamma}\}} \left|V^{\pi}_{\widehat{F},\gamma}(s) - V^{\pi}_{F,\gamma}(s)\right| \tag{18}$$

This quantity $V^{\pi}_{\widehat{F},\gamma}(s) - V^{\pi}_{F,\gamma}(s)$ is the difference between the performance of the same policy on two different environments. These two values functions should satisfy the following bellman equations:

$$V^{\pi}_{F,\gamma}(s) = \mathcal{R}(s, \pi(s)) + \lambda' F(s) + \gamma \sum_{s' \in \mathcal{S}} \mathcal{T}(s'|s, \pi(s))V^{\pi}_{F,\gamma}(s')$$

$$V^{\pi}_{\widehat{F},\gamma}(s) = \mathcal{R}(s, \pi(s)) + \lambda' \widehat{F}(s) + \gamma \sum_{s' \in \mathcal{S}} \mathcal{T}(s'|s, \pi(s))V^{\pi}_{\widehat{F},\gamma}(s')$$

To compute the solution two this equation, we use dynamic programing. Let initialize $V_0, \widehat{V} = V$(an arbitrary value) and construct the following updates.

$$for \; i \in \{1, \dots \infty\}$$

$$V^{\pi}_i(s) = \mathcal{R}(s, \pi(s)) + \lambda' F(s) + \gamma \sum_{s' \in \mathcal{S}} \mathcal{T}(s'|s, \pi(s))V^{\pi}_{i-1}(s)$$

$$\widehat{V}^{\pi}_i(s) = \mathcal{R}(s, \pi(s)) + \lambda' \widehat{F}(s) + \gamma \sum_{s' \in \mathcal{S}} \mathcal{T}(s'|s, \pi(s))\widehat{V}^{\pi}_{i-1}(s')$$

As $i$ tends to infinity, these two dynamics updates converge to $V^{\pi}_{F,\gamma}(s)$, and $V^{\pi}_{\widehat{F},\gamma}(s)$ respectively. To bound the right hand side of Eq. 18 we have

$$V^{\pi}_i(s) - \widehat{V}^{\pi}_i(s) = \lambda' F(s) - \lambda' \widehat{F}(s) + \gamma \sum_{s' \in \mathcal{S}} \mathcal{T}(s'|s, \pi(s)) \left(V_{i-1}(s') - \widehat{V}_{i-1}(s')\right)$$

$$\leq \lambda' \sum_{i'=0}^{i} \gamma^{i'} \max_s \left|F(s) - \widehat{F}(s)\right| \tag{19}$$

As $i$ tends to infinity, we have

$$\left|V^{\pi^*_{F,\gamma}}_{F,\gamma}(s) - V^{\pi^*_{\widehat{F},\gamma}}_{F,\gamma}(s)\right| = \max_\pi \left|\lim_{i \to \infty} \left(V^{\pi}_i(s) - \widehat{V}^{\pi}_i(s)\right)\right|$$

$$\leq \lambda' \sum_{i'=0}^{i} \gamma^{i'} \max_{s,\pi} \left|F(s) - \widehat{F}(s)\right| \leq \lambda' \max_{s,\pi} \frac{\left|F(s) - \widehat{F}(s)\right|}{(1 - \gamma)}$$

## B.1 LOOKUP TABLE CLASSIFIER

If we consider the fear model as a lookup table, and deterministic, then observing each state once is enough to exactly recover the classifier.

For the stochastic $F$, at time step $N$

$$\left|F(s) - \widehat{F}(s)\right| \leq \sqrt{\frac{\log \frac{N}{\delta}}{N(s)}} \tag{20}$$

with probability $\delta$ where $N(s)$ is the number visits to a state $s$ at time step $N$. The trajectory produced by algorithm does not produce $i.i.d.$ samples of state. Therefore, for Eq. 20 we use Hoeffding's inequality accompanied with union bound over time $N$. In order to have this bound to hold for all the states at once, we need another union bounds over states and all possibly optimal policies $\Pi_\gamma$ under noisy classifier , which requires to replace $\delta \to \delta / S A \Pi_\gamma$. Let's assume a minimum number of visit $\overline{N}$ to each state,

$$\|V^{\pi^*_{F,\gamma}}_{F,\gamma} - V^{\pi^*_{\widehat{F},\gamma}}_{F,\gamma}\|_\infty \le \frac{\lambda'}{1-\gamma}\sqrt{\frac{\log \frac{NSA\Pi_\gamma}{\delta}}{\overline{N}}} \tag{21}$$

Finally, adding Eq. 14 and Eq. 21, the upper bound on $\mathbb{L}$ is as follows:

$$\mathcal{L} \le \lambda' \frac{1-\gamma_{eval}}{1-\gamma}\sqrt{\frac{\log \frac{NSA\Pi_\gamma}{1-\delta}}{\overline{N}}} + \frac{\gamma_{eval}-\gamma}{1-\gamma}$$

## B.2 Classifier from a set of functions

Let $\mathcal{F}$ denote a set of given binary classifiers and $F \in \mathcal{F}$. In this case, let's assume that we are given a set of $N$ $i.i.d$ samples from the stationary distribution $\omega_F^{\pi^*_{\widehat{F},\gamma}}$ . Given a policy $\pi$, the MDP transition process reduces to a Markov chain with transition probability $\mathcal{T}^\pi$. Now we rewrite the Eq. 19 in a matrix format where $V_i^\pi, F \in \mathbb{R}^S$ are vectors of concatenation of $V_i^\pi(s)$ and $F(s)$, $\forall s \in \mathcal{S}$ respectively.

$$V_i^\pi - \widehat{V}_i^\pi = \lambda'F - \lambda'\widehat{F} + \gamma\mathcal{T}^\pi\left(V_{i-1}^\pi - \widehat{V}_{i-1}^\pi\right) \le \lambda'\sum_{i'=0}^{i}(\gamma\mathcal{T}^\pi)^{i'}\left(F - \widehat{F}\right) \tag{22}$$

as $i$ goes to infinity we have

$$V^{\pi^*_{F,\gamma}}_{F,\gamma} - V^{\pi^*_{\widehat{F},\gamma}}_{F,\gamma} \le \lambda'\left(\mathbb{1} - \gamma\mathcal{T}^\pi\right)^{-1}\left(F - \widehat{F}\right)$$

Using PAC analysis of binary classification in (Hanneke, 2016) a follow up to (Vapnik, 2013), we have

$$\left|F - \widehat{F}\right|^\top \omega_F^{\pi^*_{\widehat{F},\gamma}} \le 3200\frac{\mathcal{VC}(\mathcal{F}) + \log\frac{1}{\delta}}{N}$$

with probability at least $1 - \delta$ where $\mathcal{VC}(\mathcal{F})$ is the $\mathcal{VC}$ dimension of the hypothesis class and $|\cdot|$ is entry-wise absolute value. Since $\gamma < 1$, then $\alpha_{max}$, the maximum eigenvalue of $(\mathbb{1} - \gamma\mathcal{T}^\pi)^{-1}$, is bounded above and we have

$$\|V^{\pi^*_{F,\gamma}}_{F,\gamma} - V^{\pi^*_{\widehat{F},\gamma}}_{F,\gamma}\|_1 \le 3200\lambda'\alpha_{max}\frac{\mathcal{VC}(\mathcal{F}) + \log\frac{1}{\delta}}{N}$$

and therefore,

$$\mathcal{L} \le 3200\lambda'\alpha_{max}(1 - \gamma_{eval})\frac{\mathcal{VC}(\mathcal{F}) + \log\frac{1}{\delta}}{N} + \frac{\gamma_{eval}-\gamma}{1-\gamma}$$

The remaining part is to solve

$$\gamma^* = \text{argmin}_{\gamma \le \gamma_{eval}}\mathcal{L}$$

to find the optimal $\gamma$.

The same analysis, up to a slight modification[3], holds for the continuous state and action spaces.

---

[3]Instead of having $V$ as a vector of state values indexed by states, it is a continuous function of states. Furthermore, the Transition kernel is over continuous distribution therefore the same bellman update in Eq. 22 holds.

