# OpenReview forum: "Avoiding Catastrophic States with Intrinsic Fear"
_ICLR.cc/2018/Conference — Reject_

### Official Review · AnonReviewer3 · 2017-11-17
**There could be many other base-line ideas that could avoid the catastrophic scenarios.**

**Rating:** 5
**Confidence:** 5

**Review:**


SUMMARY

The paper proposes an RL algorithm that combines the DQN algorithm with a fear model.  The fear model is trained in parallel to predict catastrophic states.  Its output is used to penalize the Q learning target.



COMMENTS

Not convinced about the fact that an agent forgets about catastrophic states. Because it does not experience it any more.  Shouldn’t the agent stop learning at some point in time?  Why does it need to keep collecting good data?  How about giving more weight to catastrophic data (e.g., replicating it)

Is the catastrophic scenario specific to DRL or RL in general with function approximation?

Why not specify catastrophic states with a large negative reward?

It seems that catastrophe states need to be experienced at least once.
Is that acceptable for the autonomous car hitting a pedestrian?

---

> ### Author Response · Authors · 2018-01-04
> **Response to AnonReviewer3**
>
> We are grateful to Reviewer3 for taking the time to review our paper but disagree with several of the assertions.
>
> 1. The reviewer states “Not convinced about the fact that an agent forgets about catastrophic states”. The susceptibility of neural networks to catastrophic forgetting (not to be confused with our safety-motivated notion of a catastrophe) is well-documented in the literature. Whenever the policy is modified such that some states would never be encountered, they will eventually, as soon as they are flushed from the replay buffer, cease to influence the Q-network. If we continue to update the network as is necessary, especially in non-stationary environments (e.g. nearly all real-world settings) then nothing in the standard DQN formulation guards the agent from revisiting the catastrophic states.
>
> In addition to being well-documented in the literature, we demonstrate this problem clearly in our paper using the simplest failure case. Even in AdventureSeeker, a 1-D environment with only two actions, the agent will eventually forget about the catastrophic states.
>
> 2.  Re: “Shouldn’t the agent stop learning at some point in time?”
>
> a. First, even when there is limited duration learning period, we may want an agent to make a minimal number of catastrophic errors while learning.
>
> b. Second, as stated above: in nonstationary environments, which describes most real-world environments, we want to learn continually.  Otherwise the policy will become stale and cease to perform owing to the shifting dynamics. In the case of driving, imagine new cars which appear on the road, or new street signs. In the case of a vacuum-cleaner, imagine that it is confronted with new household appliances that didn’t exist previously.
>
> 3. “Re: “Why not specify catastrophic states with a large negative reward?”
>
> Unfortunately, even large negative rewards are eventually forgotten, leading the agent to revisit the catastrophic states.  Take AdventureSeeker as an example: no matter how negative the penalty is, the same catastrophic forgetting will eventually happen. Moreover, this approach, unlike ours, has no notion of “danger zone” and therefore does not benefit from reward shaping. In our approach, the agent avoids even getting *close* to a catastrophe. When this assumption is reasonable, this leads to significantly faster exploration.
>
> 4. Re:  “It seems that catastrophe states need to be experienced at least once. Is that acceptable for the autonomous car hitting a pedestrian?”
>
> This is a good question that has implications for all of RL: If catastrophes can truly never be experienced even once, then is reinforcement learning off the table altogether?
>
> However, in many settings, perhaps even car accidents, if enough cars are on the road and the probability of an accident is nonzero, then accidents will happen. Our work addresses how to learn from these mistakes rapidly and to guard against repeating the same mistakes in the future.

---

### Official Review · AnonReviewer2 · 2017-11-27
**Interesting idea, but some potential issues.**

**Rating:** 5
**Confidence:** 4

**Review:**

The paper addresses the problem of learners forgetting rare states and revisiting catastrophic danger states. The authors propose to train a predictive ‘fear model’ that penalizes states that lead to catastrophes. The proposed technique is validated both empirically and theoretically.

Experiments show a clear advantage during learning when compared with a vanilla DQN. Nonetheless, there are some criticisms than can be made of both the method and the evaluations:

The fear radius threshold k_r seems to add yet another hyperparameter that needs tuning. Judging from the description of the experiments this parameter is important to the performance of the method and needs to be set experimentally. There seems to be no way of a priori determine a good distance as there is no way to know in advance when a catastrophe becomes unavoidable. No empirical results on the effect of the parameter are given.

The experimental results support the claim that this technique helps to avoid catastrophic states during initial learning.The paper however, also claims to address the longer term problem of revisiting these states once the learner forgets about them, since they are no longer part of the data generated by (close to) optimal policies.  This problem does not seem to be really solved by this method. Danger and safe state replay memories are kept, but are only used to train the catastrophe classifier. While the catastrophe classifier can be seen as an additional external memory, it seems that the learner will still drift away from the optimal policy and then need to be reminded by the classifier through penalties. As such the method wouldn’t prevent catastrophic forgetting, it would just prevent the worst consequences by penalizing the agent before it reaches a danger state. It would therefore  be interesting to see some long running experiments and analyse how often catastrophic states (or those close to them) are visited.

Overall, the current evaluations focus on performance and give little insight into the behaviour of the method. The paper also does not compare to any other techniques that attempt to deal with catastrophic forgetting and/or the changing state distribution ([1,2]).

In general the explanations in the paper often often use confusing and  imprecise language, even in formal derivations, e.g.  ‘if the fear model reaches arbitrarily high accuracy’ or ‘if the probability is negligible’.

It is wasn’t clear to me that the properties described in Theorem 1 actually hold. The motivation in the appendix is very informal and no clear derivation is provided. The authors seem to indicate that a minimal return can be guaranteed because the optimal policy spends a maximum of epsilon amount of time in the catastrophic states and the alternative policy simply avoids these states. However, as the alternative policy is learnt on a different reward, it can have a very different state distribution, even for the non-catastrophics states.  It might attach all its weight to a very poor reward state in an effort to avoid the catastrophe penalty. It is therefore not clear to me that any claims can be made about its performance without additional assumptions.

It seems that one could construct a counterexample using a 3-state chain problem (no_reward,danger, goal) where the only way to get to the single goal state is to incur a small risk of visiting the danger state. Any optimal policy would therefore need to spend some time e in the danger state, on average. A policy that learns to avoid the danger state would then also be unable to reach the goal state and receive rewards. E.g pi* has stationary distribution (0,e,1-e) and return 0*0+e*Rmin + (1-e)*Rmax. By adding a sufficiently high penalty, policy pi~ can learn to avoid the catastrophic state with distribution (1,0,0) and then gets return 1*0+ 0*Rmin+0*Rmax= 0 < n*_M - e (Rmax - Rmin) = e*Rmin + (1-e)*Rmax - e (Rmax - Rmin).  This seems to contradict the theorem. It wasn’t clear what assumptions the authors make to exclude situations like this.

[1] T. de Bruin, J. Kober, K. Tuyls and R. Babuška, "Improved deep reinforcement learning for robotics through distribution-based experience retention," 2016 IEEE/RSJ International Conference on Intelligent Robots and Systems (IROS), Daejeon, 2016, pp. 3947-3952.
[2] Kirkpatrick, J., Pascanu, R., Rabinowitz, N., Veness, J., Desjardins, G., Rusu, A. A., ... & Hassabis, D. (2017). Overcoming catastrophic forgetting in neural networks. Proceedings of the National Academy of Sciences, 201611835.

---

> ### Author Response · Authors · 2018-01-04
> **Response to AnonReviewer2**
>
> Thanks for the thoughtful review of our paper.
>
> 1. We are glad that you noticed the issue in the proof of theorem 1. Per your feedback, we have corrected the proof and substantially revised the paper (see current revision). At a high level, the performance degradation, as described corrected theorem and proof are as follows:
>
> If the optimal policy, \pi^*, of the original environment, without intrinsic fear, (M), visits the fear zone with probability at most \epsilon, then applying  pi^* on the environment with intrinsic fear (M,F), gives the return of eta^*-\epsilon\lambda(Rmax-Rmin) therefore, the optimal policy on (M,F), \tilde{\pi}, can not give a return less than \eta^*\epsilon-\lambda(Rmax-Rmin) on environment (M,F). If \tilde{\pi} visits the fear zone with probability \epsilon’, we can rewrite its return as:
> (return from non intrinsic fears)-epsilon’(\lambda(Rmax-Rmin)
> Therefore applying \tilde{\pi} on original environment (M) gives a return of at least \eta^*-\epsilon\lambda(Rmax-Rmin) +\epsilon’\lambda(Rmax-Rmin) which is lower bounded by \eta^*-\epsilon\lambda(Rmax-Rmin).
>
> 2. Regarding the parameter k_r, as the reviewer mentioned, without any prior knowledge and posed safety constraint of the environment, this parameter needs to be chosen empirically, as with other hyper-parameters. We note however, that this is a kind of prior knowledge that might be reasonably to expect of an algorithm designer. For example, a robot should perhaps never be too close to a cliff or a ledge.
>
> Intuitively, small k_r’s better preserve the original policy, but for too small a k_r, the fear model might be ignored. On the other hand, large k_r are better at preventing the agent from visiting the catastrophic states but run more risk of deviating substantially from the optimal policy. Prior knowledge of the environment can guide us to design a proper k_r, otherwise, k_r needs to be chosen experimentally.
>
> 3. Regarding the (very) long term forgetting, the reviewer is correct that this paper doesn’t completely alleviate catastrophic forgetting an that we instead guard against the worst consequences. We have created a video to visualize the fear probability as a red overlay on the video game play and will continue to work on other ways to qualitatively understand how our algorithm is working.
>
> 4. We thank the reviewer for suggesting baselines to compare to.  They have some relevance but are designed for different purposes.  In particular,
> [1] (IROS) uses a second experience replay buffer to store state transitions that covers the whole state space uniformly, in addition to a typical buffer used in standard DQN.  This approach aims mostly to reduce exploration, but can face the curse of dimensionality as it tries to cover the state space uniformly.  Moreover, the uniform covering idea is not efficient for avoiding catastrophic events that are rare, while our approach uses a fear classifier to target danger zones directly.
> [2] (PNAS) takes a Bayesian approach to continual learning, trying to avoid catastrophic forgetting of solutions to earlier tasks that have not occured for a long time.  In contrast, our problem is to avoid running into catastrophic states in the same task.  It is not clear how a similar, Bayesian variant of DQN (such as BBQ) can be extended to address our safe exploration challenge.

---

> ### Comment · AnonReviewer2 · 2018-01-15
> **Updated score**
>
> I've slightly increased my score to reflect the improvements made by the authors. Theorem 1 seems to have been corrected. Unfortunately, the bound now indicates that the average reward is within lambda * epsilon * (R_max - R_min) of the optimal average reward (where lambda can be arbitrarily large). This does not provide much in the way of guarantees.
>
> My final feeling about the paper is that it introduces a mostly heuristic method, which can provide some empirical benefit when properly tuned. It wasn't clear to me, however, that this fear model offers a generic method or that is capable of achieving the goals the authors mention.

---

### Official Review · AnonReviewer1 · 2017-11-30
**DQN and catastrophic forgetting**

**Rating:** 7
**Confidence:** 3

**Review:**

The paper studies catastrophic forgetting, which is an important aspect of deep reinforcement learning (RL). The problem formulation is connected to safe RL, but the emphasis is on tasks where a DQN is able to learn to avoid catastrophic events as long as it avoids forgetting. The proposed method is novel, but perhaps the most interesting aspect of this paper is that they demonstrate that “DQNs  are susceptible to periodically repeating mistakes”. I believe this observation, though not entirely novel, will inspire many researchers to study catastrophic forgetting and propose improved strategies for handling these issues.

The paper is accurate, very well written (apart from a small number of grammatical mistakes) and contains appealing motivations to its key contributions. In particular, I find the basic of idea of introducing a component that represents fear natural, promising and novel.

Still, many of the design choices appear quite arbitrary and can most likely be improved upon. In fact, it is not difficult to design examples for which the proposed algorithm would be far from optimal. Instead I view the proposed techniques mostly as useful inspiration for future papers to build on. As a source of inspiration, I believe that this paper will be of considerable importance and I think many people in our community will read it with great interest. The theoretical results regarding the properties of the proposed algorithm are also relevant, and points out some of its benefits, though I do not view the results as particularly strong.

To conclude, the submitted manuscript contains novel observations and results and is likely to draw additional attention to an important aspect of deep reinforcement learning. A potential weakness with the paper is that the proposed strategies appear to be simple to improve upon and that they have not convinced me that they would yield good performance on a wider set of problems.

---

> ### Author Response · Authors · 2018-01-04
> **Response to AnonReviewer1**
>
> We thank AnonReviewer1 for a clear and constructive review. We are encouraged that you recognize the importance of the problem addressed and the novelty of the methods. Per your suggestions, we have polished the paper, fixing several of the typos that had made it into the first draft. The reviewer’s point that many aspects of the algorithm can likely be improved upon in future work is well-taken. We hope that this is just one of the first among many papers to improve with respect to these fundamental problems.

---

### Public Comment · ~Nick_Linck1 · 2017-11-16
**Source Code**

I was wondering if you have the code open sourced so that we can more easily reproduce the results provided in the paper.

---

> ### Author Response · Authors · 2017-11-19
> **Posted Code**
>
> Hi Nick,
>
> Thanks for your interest in our paper! The code is actually open sourced now, and already one group of researchers has re-implemented our algorithm from scratch and confirmed outperformance of DQN. To preserve double blind status, we won't post the GitHub link here but it's not too hard to find.
>
> Cheers,
>
> Authors

---

### Public Comment · ~Zhe_Du1 · 2017-11-22
**Post Parameters?**

Hi,

This is a nice paper and we like the ideas in it! I tried to implement the algorithm DQN-Fear by modifying the baseline DQN, and reproduce the simulation results in your paper. The thing is, based on our trials with different parameters so far, we have some difficulty to reproduce part of the result. For example, (1) for the CartPole test, the DQN runs more than 10000 episodes within 4e6 time steps, while in your paper, there are only 4000 episodes;  (2) (this is more weird) for Freeway, our DQN achieves better performance than the plot of DQN-Fear in your paper within just 300 episodes. I guess this may be largely due to the hyper parameters.

We really appreciate your code on GitHub, and we can see the parameters for Atari games. But the hyperparameters for Adventure seeker and Cartpole are still unclear.  So, I am wondering if it's possible you share the hyper-parameters for DQN and DQN-Fear on all three experiments? The hyper-paramters could include but not limit to the following:
(1) AdamLearning rates for the two neural nets of DQN and fear model
(2) Buffer sizes for all 3 buffers
(3) How exploration rate is scheduled
(4) Train frequency
(5) Batch size
(6) When do the the learning start for the two neural nets of DQN and fear model
(7) discount factor gamma
(8) Target network update frequency
(9) fear factor
(10) fear phase-in length
(11) fear radius

Thank you!

---

> ### Author Response · Authors · 2017-11-27
> **Of course!**
>
> Happy to share details - pardon the delay due to holiday travel. Need to get back home to look up exact details on hyperparameter settings as the toy environment experiments were done a while ago.
>
> Yes the hyper-parameters can make a big difference on many of these problems. Optimizer, number of exploration turns, etc. There's also a large amount of variance across runs. Especially on the toy environments. That's why we run every experiment multiple times and report averages.
>
> Thanks for the questions and for holding tight, more details on toy environments coming soon!

---

### Public Comment · ~Jonathan_Adalin1 · 2017-12-16
**Evaluating the Reproducibility of the Intrinsic Fear Model**

The goal of this summarized review is to investigate the reproducibility of the results found in the above report on the use of instrinsic fear in the context of reinforcement learning. In doing so, the assessment aims to contribute to the machine learning community by allowing others who wish to make use of the latter research to do so with confidence and ease. A link to the full report can be found at the end of this thread.

The authors of "Avoiding Catastrophic States with Intrinsic Fear" claim that incorporating the distancing from dangerous scenarios into the reward system of the DQN model will shorten training time and reduce the number of catastrophes that the learning agent will put itself in. The latter is demonstrated using toy environments Adventure-Seeker and CartPole, as well as in Atari games Seaquest, Asteroids and Freeway. The two models attempt to learn in these environments and their results are compared empirically.

To evaluate the reproducibility of these findings, we chose two of the five environments to train the learning agents. The reason all five were not selected were because of the time constraint of this assignment as well as the limited hardware that our team had in possession. To train the agents, we used a combination of open-source code and self implemented models. The empirical results that were found showed that the intrinsic fear model did in fact outperform the standard DQN model as the authors of the original report suggested.

In order to evaluate the ease of reproducibility of this research paper, we evaluated the latter under four metrics:

1) Availability of code, names and version numbers of dependencies

The code from the authors' was not explicitly given to us, but they anonymously hinted that their code was open sourced on Github. After using this code to reproduce the Asteroids scenario, we were able to refer to the code to implement an intrinsic fear for the CartPole environment. Unfortunately, the version numbers of the dependencies were not explicitly given but through research we managed to figure out the preliminary steps required to run the authors' code.

2) Clarity of code and paper

Hyper-parameters in the code were specified in a single section with appropriate descriptions. Additionally, the concepts specific to the report were well explained with approachable examples.

3) Details of computing infrastructure used and computation requirements

No information was given relating the computing infrastructure, so we were unable to know if our hardware was sufficient to fully train the agents without running the learners first.

4) Reimplementation effort

Considering the limited time given for this assignment, the fact that we were able to successfully reproduce the findings in the original paper is an strong indication that the latter is reproducible.

After considering said criteria, it became clear that "Avoiding Catastrophic States with Intrinsic Fear" was a reproducible report and we must applaud the authors in their ability to convey such a complex topic in an approachable and duplicable manner.

The full report on reproducibility can be found here:
https://drive.google.com/open?id=1QSEIgg2f22Cd06mA-23IthE-9M9o7Kpe

---

### Public Comment · ~Ksenia_Kolosova1 · 2017-12-16
**Replication study of Intrinsic Fear paper**

Being part of the ICLR 2018 Reproducibility Challenge, we worked to reproduce the results presented in this paper (Avoiding Catastrophic States with Intrinsic Fear) currently under a double-blind review process at the time of writing of this comment. We enjoyed reading the paper and replicating their results.

	This paper proposed a model for intrinsic fear, a reward shaping model that improves the functioning of a deep-Q network by minimizing the number of catastrophes experienced during training. The authors tested their model on the popular reinforcement learning game Cartpole, their own game Adventure Seeker, and the three Atari games Seaquest, Asteroids, and Freeway.

	In this paper, they only showed detailed plots for reward and experienced catastrophes for the Atari games. As such, we set out to replicate their results for those games. The code was available online in a Github repository, as directed by the authors in a comment to another member of the public. We had some trouble running the code at first as there were no guidelines for the usage of the code such as package versions and necessary dependencies, but we were able to adapt it for our environment. Our slightly adapted version of their code, and a README file that details how we ran their code, can be found along with our full report linked at the end of this comment.

In our runs of their code, we found that on Freeway, we were able to reproduce better performance of the DQN with intrinsic fear (DQN-IF) over that of a normal DQN without intrinsic fear, in terms of the reward gained. Our plot did not look exactly like the one in the paper, which may be due to the authors’ averaging their results over multiple runs of the same experiment. It was not mentioned in the original report but pointed out by the authors in the OpenReview discussion that there is averaging of the total rewards per episode over many learning runs for the Atari games. It may also be due to differences in hyperparameter settings that were not made explicit in the paper. However, our result for the catastrophe rate for Freeway did not match the result in the paper. In fact, we found the catastrophe rate looked identical with and without intrinsic fear, which may also be due to different hyperparameter usage. Another interesting observation from our runs on Freeway was that identical hyperparameter settings led to different results on our two different systems, suggesting that library versions or hardware may play a role in the performance of the algorithm.

On Asteroids, we found that at approximately 4000 episodes we were not able to prove the dominance of DQN-IF over DQN in terms of obtained reward. After one run of each model, the variation in reward for training was too high to observe the clear trend that was presented in the paper after averaging, though qualitatively the DQN-IF experiences marginally higher reward. The total reward however was much lower than that found in the paper for both models. Nevertheless, our results showed that the DQN model visited more catastrophe states than the DQN-IF within the same timeframe, which reflects the data presented in the original paper.

Our results for Seaquest were insufficient to draw any conclusions due to our limited computational resources, but the results for the obtained reward over 3000 episodes looked similar to trends observed in the paper. Interestingly, the decrease in catastrophe rate for the DQN-IF compared to the DQN seemed to show a promising trend.

	To summarize, we assume that the differences between our results and those presented in the paper are due in part to different hyperparameter settings. Another potential reason is the fact that we were not able to do enough runs to average results. Unfortunately, we do not know the exact averaging conditions, so our graphs are not as smooth as the ones presented in the paper.

	Some features that would have been useful to us for our replication are (1) clearer definition of hyperparameters for all games, including the missing fear factor for CartPole (2) figures of results for Cartpole and Adventure Seeker (3) online availability of Adventure Seeker (4) some discussion of computational resources required (5) a usage guide for the code, as well as improved saving and restart functionality (6) explanation for hyperparameter selection and optimization to aid efforts to implement this model in other learning environments.

Our full report and code can be found here: https://drive.google.com/drive/folders/1DSsQq4YRiwA-KpEancIo-GAvsPYQ-u4K

---

> ### Public Comment · (anonymous) · 2017-12-28
> **Ditto to Questions about Reproducibility**
>
> After reading the paper I also had some questions about the DQN baseline used in the paper. After running some simple experiments with DQN on Freeway, it seemed to me that the results reported by the paper for DQN in this paper were underpowered, as an out-of-the-box DQN got superior results. I came on here to comment this but just saw the above post so I'll just leave this here as a reply (thanks to the commenter above for their thorough experiments).

---

> > ### Author Response · Authors · 2018-01-04
> > **Reply**
> >
> > Note that there are many out-of-the-box DQNs available. They do not all achieve the same performance on every game. DRL is unfortunately still rather brittle to small implementation changes. For example, if you alter (SGD vs Momentum vs ADAM), size of initial replay buffer before reducing epsilon to < 1, etc. you will notice that often each agent will do better for some games (sometimes strikingly) and worse on others. We cannot vouch for the performance of every configuration of DQN you might access, only for the specific implementation that we used. One small detail that could potentially explain some differences is that we used a smaller initial replay buffer size than in the original DQN paper. Perhaps this additional early exploration was crucial for DQN but not for the Intrinsic Fear model.

---

### Author Response · Authors · 2018-01-04
**General reply to reviewers and area chair**

We would like to thank the reviewers for taking the time to leave thoughtful reviews. Given this feedback, we have significantly improved the draft and hope the reviewers and area chair will take this into account when assessing the final scores. For example, the harsh score from reviewer 2 owes largely to a mistake in one theorem that has since been fixed in the newest version. We are also grateful to the folks at the reproducibility project who noted the commendable clarity and reproducibility of our paper, algorithm and empirical findings. Please find specific rebuttals to each reviewer as replies to the respective reviews.

---

### Decision · Program_Chairs · 2018-01-29
**ICLR 2018 Conference Acceptance Decision**

**Decision:**

Reject

**Comment:**

This paper presents an interesting idea that is related to imitation learning, safe exploration,
and intrinsic motivation. However, in its current state the paper needs improvement in clarity. There are also some concerns about the number of hyperparameters involved. Finally, the experimental results are not completely convincing and should reflect existing baselines in one of the areas described above.